# Prognostic Factors and Current Treatment Strategies for Renal Cell Carcinoma Metastatic to the Brain: An Overview

**DOI:** 10.3390/cancers13092114

**Published:** 2021-04-27

**Authors:** Valeria Internò, Pierluigi De Santis, Luigia Stefania Stucci, Roberta Rudà, Marco Tucci, Riccardo Soffietti, Camillo Porta

**Affiliations:** 1Department of Biomedical Sciences and Human Oncology, University of Bari, 70122 Bari, Italy; p.desantis@studenti.uniba.it (P.D.S.); stefania.stucci@uniba.it (L.S.S.); marco.tucci@uniba.it (M.T.); camillo.porta@uniba.it (C.P.); 2Aldo Moro Division of Medical Oncology, A.O.U. Consorziale Policlinico di Bari, 70121 Bari, Italy; 3Department of Neurology, Castelfranco Veneto and Treviso Hospital, 31033 Castelfranco Veneto, Italy; roberta.ruda@aulss2.veneto.it; 4Department of Neuro-Oncology, University and City of Health and Science Hospital, 10122 Turin, Italy; riccardo.soffietti@unito.it; 5National Cancer Research Center, Tumori Institute IRCCS Giovanni Paolo II, 70121 Bari, Italy

**Keywords:** renal cell carcinoma, brain metastasis, prognostic factors, cytoreductive nephrectomy, immunotherapy, target therapy, stereotactic radiosurgery, combined treatment, radiation therapy

## Abstract

**Simple Summary:**

Brain metastases are a commonly recognized poor prognostic factor in patients with cancer. Due to their poor prognosis, these patients were commonly excluded from the most important clinical trials that have revolutionized the oncological clinical practice. Renal cell carcinoma represents one of the most frequent neoplasia that metastasize to the brain. Due to the therapeutical advances the overall survival of brain metastatic renal cell carcinoma is improved even in the absence of tailored studies that are need to plan an adequate therapeutic strategy for these patients.

**Abstract:**

Renal cell carcinoma (RCC) is one of primary cancers that frequently metastasize to the brain. Brain metastasis derived from RCC has the propensity of intratumoral hemorrhage and relatively massive surrounding edema. Moreover, it confers a grim prognosis in a great percentage of cases with a median overall survical (mOS) around 10 months. The well-recognized prognostic factors for brain metastatic renal cell carcinoma (BMRCC) are Karnofsky Performance Status (KPS), the number of brain metastasis (BM), the presence of a sarcomatoid component and the presence of extracranial metastasis. Therapeutic strategies are multimodal and include surgical resection, radiotherapy, such as stereotactic radiosurgery due to the radioresistance of RCC and systemic strategies with tyrosin kinase inhibitors (TKI) or Immune checkpoint inhibitors (ICI) whose efficacy is not well-established in this setting of patients due to their exclusion from most clinical trials. To date, in case of positive prognostic factors and after performing local radical therapies, such as complete resection of BM or stereotactic radiosurgery (SRS), the outcome of these patients significantly improves, up to 33 months in some patients. As a consequence, tailored clinical trials designed for BMRCC are needed to define the correct treatment strategy even in this poor prognostic subgroup of patients.

## 1. Introduction

Renal cell carcinoma (RCC) accounts for 3–5% of all solid malignancies diagnosed worldwide [1]. Approximately one-third of patients with RCC present with metastatic disease at diagnosis, and amongst those with a radically resected localized disease, around 30% will develop metachronous metastases, in up to 17% of cases localized to the brain [2,3].

Patients with sarcomatoid component, large tumor size and lymph node involvement have a higher risk of developing brain metastases (BM) [4]. Despite the improved outcome of metastatic RCC (mRCC) patients, mainly due to the availability novel systemic therapies, BM are associated with a worse prognosis with a median overall survival (mOS), hardly reaching 10 months [5,6,7]. Poor prognosis might be related to the typical resistance to radiotherapy of brain metastatic renal cell carcinoma (BMRCC), the poor penetration of blood-brain barrier (BBB) by many anti-cancer agents [8,9,10], and also by the frequent occurrence of symptomatic neurological impairment. Indeed, central nervous system (CNS) symptoms are observed in as much as 70–80% of the patients with BMRCC.

Furthermore, BMRCC display a higher risk of intracranial hemorrhage due to its hypervascular structure [11], often leading to the need for surgical intervention to manage neurological complications [5,12].

To our knowledge, there are no officially accepted guidelines regarding CNS surveillance in RCC patients, neither in the non-metastatic, nor in metastatic, settings. Although, data are available regarding the improved outcome in well-selected patients due to early BM diagnosis [13,14].

In relation to BMRCC treatment, neuro-surgery or stereotactic radiosurgery (SRS) alone, or in combination, represent the gold standard in single or oligo-metastasis. While, whole brain radiotherapy (WBRT) progressively lost its role, and thus, it is limited to the treatment of multiple metastases not amenable to focal therapy [15].

New treatment options, based on targeted agents and/or immunotherapy, have dramatically improved the outcome of metastatic RCC (mRCC) patients. On the other hand, no general consensus on the activity and efficacy of these treatment in BMRCC exist, due to the limited evidence available and the consequent lack of consensus among the scientific community [16].

In this review, we will discuss BMRCC’s prognostic factors and current treatment strategies, particularly focusing on BM response, thus, realizing a comprehensive overview of therapeutic efficacy and outcome in this setting of patients, often excluded from clinical trials.

## 2. Clinico-Radiological Features of BMRCC and Indications for Brain Surveillance in RCC Patients

BM occur in mRCC with an incidence ranging from 2% to 17% [3,17,18,19], which appears to have significantly increased in the last two decades as a consequence of the increasing life expectancy of mRCC patients due to the availability of effective systemic therapies [4].

An estimated 2.4% of patients with non-metastatic RCC develops metachronous BM, while only 6.5% have BM already at the time of primary diagnosis [2]. Therefore, BMRCC more often occurs as a metachronous dissemination through different pathways, the most relevant being the cava-type pathway (75% of the cases, Figure 1). This justifies the common observation that BMRCC patients usually present also lung metastasis [2,20,21].

To date, several published studies found an increased risk of developing BMRCC in patients with clear cell histology, sarcomatoid differentiation, younger age (<70 years), larger tumour size (>7 cm or 10 cm) and lymphnode metastatic involvement [4,22,23].

In relation to the radiological features of BMRCC, these lesions are usually characterized by an increased vascularity, often associated with significant vasogenic peritumoral oedema, with a consequent higher risk of intracranial hemorrhage. Overall, this leads to localized and non-localized CNS symptoms, such as headache, confusion, altered mental status or behavior, sensor and/or motor deficits, as well as seizures [3,24,25,26,27].

If untreated, asymptomatic patients will usually become symptomatic with time [28].

At the time of our analysis, current guidelines for localized RCC do not recommend brain imaging evaluation unless patients present CNS symptoms [1], even if published guidelines of the National Comprehensive Cancer Network (NCCN) recommend brain imaging surveillance in other tumors with brain tropism, such as lung cancer, breast cancer and melanoma, even in asymptomatic patients. This type of brain surveillance has led to improved OS, which was attributed to early detection and treatment [29].

In relation to RCC, some retrospective studies have shown potentially survival benefits derived from CNS screening, with the aim of early identifying smaller lesions that are more amenable to less invasive treatments, potentially resulting in decreased morbidity. In this regard, the evidence of solitary and smaller BM in selected younger patients with a good performance status can lead to a prolonged survival from aggressive therapy, and decreased CNS recurrence risk. [5,30].

As a consequence, periodic CNS surveillance during the treatment of mRCC, without BM, may be worth doing, in order to identify patients who can benefit from earlier therapy aimed at survival improvement [31].

## 3. BMRCC’s Prognostic Factors and Risk Scales

Several researchers looked at BMRCC clinico-molecular and radiological features with respect to their outcome, with the aim of identifying prognostic factors and developing risk scales to guide clinicians’ therapeutic choices.

In this regard, in 2010, Sperduto et al. published a multi-institutional analysis of 4259 patients with BM from solid tumors. This led to the development of the Diagnosis-Specific Graded Prognostic Assessment (DS-GPA) tool, aimed at identifying significant diagnosis-specific prognostic factors and indexes. The hypothesis of their analysis was based on the concept that BM might behave differently depending on the primary tumor type, and as a consequence, should be treated differently based on their primitive site of disease. In this analysis, the authors found that significant prognostic factors varied with histological diagnosis and, focusing on RCC, identified as significant prognostic factors the Karnofsky performance status and the number of BMs (1 versus 2–3 versus >3) [32].

Further studies specifically focused on BMRCC. In particular, Bennani et al. retrospectively analyzed prognostic factors of 28 patients with BMRCC. mOS from BM onset was 13.3 months, one-year survival rate was 60.2% and two-year survival rate was 16.4%. Significant prognostic factors since BM’s diagnosis were the presence of intracranial hypertension (ICH), and/or other systemic metastasis, the absence of deep BMs metastasis and the surgical resection, the latter underlying the importance of defining surgical criteria for operability. In fact, patients who underwent surgery exhibited a markedly longer survival (25.7 months). Whereas, non-operated patients had a shorter survival (8.5 months). The surgical selection of patients has been influenced by the presence of a single brain metastasis, an accessible location within the brain, as well as a controlled systemic disease. The authors, in this analysis, did not confirm the prognostic role of DS-GPA criteria, evidencing a non-uniformity of accepted BMRCC prognostic factors [13].

Moreover, poor MSKCC risk score, the presence of sarcomatoid component and more than 3 BMs were indicated as prognostic factors of poorer outcome in a multivariate analysis of 93 BMRCC patients treated at University of Ulsan College of Medicine. On the other hand, local treatment was identified as an independent factor for a better OS. In this work, which was published in 2017, Choi SY et al. examined the survival differences between synchronous and metachronous BM, and did not find any differences, in terms of BM progression and OS after the diagnosis of BM [14].

Takeshita et al. confirmed, in their study, which was published in 2019, the prognostic factors evidenced in the above-mentioned papers, including DS-GPA. In particular, in their study, BMRCC features are associated with poor OS were KPS < 70, a DS-GPA score ≤ 2, no treatment for brain metastasis and the presence of sarcomatoid components. Moreover, OS plots showed a lower mOS for patients with a GPA ≤ 2 than those with a GPA > 2 as for patients with sarcomatoid components [33].

More recently, El Ali et al. retrospectively analysed a cohort of 93 BMRCC patients in order to validate, in mRCC, two prognostic scores formulated for BM from different tumor hystologies (namely the “Radiation Therapy Oncology Group Recursive Partitioning Analysis”—RTOG RPA and the “Basic Score for Brain Metastases”—BS-BM). Moreover, the same authors formulated a potential new prognostic score, named CERENAL score, which is specifically tailored on RCC. In this work, patients were distributed among RTOG RPA classes I, II or III basing on age, KPS and presence of extracranial metastases at brain metastasis diagnosis. On the other hand, the BS-BM was determined using KPS, management of systemic disease and presence of extracranial metastases at brain metastasis diagnosis.

The new edited CERENAL prognostic score was based on the prognostic parameters used for the RTOG RPA and the BS-BM and on new findings that showed that a low number of intracranial metastases and SRS may predict a substantial survival benefit for metastatic RCC patients. In their analysis, all prognostic scores showed significance for PFS after first-line targeted therapy and a multivariate analysis proved that the CERENAL score was the sole independent prognostic factor associated with an improved PFS from first-line therapy. In relation to OS, the authors confirmed, in a univariate analysis the prognostic value of the three prognostic scores [34].

The principal characteristics of BMRCC prognostic score are summed in Table 1.

## 4. Role of Cytoreductive Nephrectomy

Beyond the debate which surrounds cytoreductive nephrectomy (cyNx) in the post-CARMENA era, clinicians usually do not suggest cyNx in patients with BMRCC, due to their poor prognosis (mOS 11 months), especially when lesions are synchronous. Although no prospective data are available in this setting, to date, BMRCCs are considered a de facto contraindication to cyNx.

Despite all the above, Daugherty M. et al. retrospectively collected clinico-therapeutic information and outcomes of 775 synchronous BM from RCC, focusing on the putative prognostic role of cyNx. In their cohort, the authors evidenced that patients with brain-only metastasis were more likely to undergo cyNx with respect to the counterpart with extracranial metastasis (40.8% vs. 20.8%). More importantly, patients with isolated BM who underwent cyNx displayed similar survival rates than those with isolated lung, liver, or bone metastasis, thus, showing that the presence of isolated BM did not correlate with a worse prognosis, as compared to another isolated extracranial metastasis. Similarly, in patients who did not undergo cyNx, no significant differences, in terms of outcome based on the site of isolated metastases (brain only versus lung only, brain only versus liver only, and brain only versus bone only) were observed. Furthermore, the subgroup with isolated BMRCC showed a better outcome if compared to those with multiple sites of disease with a 1 and 2 year survival of 44%, and 31%, respectively, for the brain-only subgroup. Notably, patients who underwent cyNx in the setting of brain-only metastasis had a significantly improved one- and two-year survival of 67%, and 52%, respectively, and an overall median survival of 33 months. Interestingly, patients with isolated BMRCC, who were not treated with cyNx, had a one- and two-year survival rates of 26% and 14%, and a median survival of five months. From these results, the detection of BMRCC should not necessarily be considered an indication of poor prognosis; what seems more important is the presence of an oligometastatic disease, and in particular, one metastatic site only, irrespective of the anatomic localization [35].

Zhuang et al. examined 933 RCC patients diagnosed between 2010 and 2014 with BM within the Surveillance, Epidemiology and End Results (SEER) database. The authors found that cyNx provided significant benefits in terms of survival. Indeed, survival analysis performed on matched cases showed that mOS of nephrectomized patients was 14.0 months, versus 5.0 months for the non-surgical group, a difference which was statistically significant. To date, it has been taken in count that cyNx has a 30-day mortality of 0.5–1.8% and that the perioperative mortality is associated with increasing tumour stage and depends also on surgeon and hospital operative volume. Moreover, severe complications occur in 3–8% of patients undergoing cyNx [22].

As a whole, prospective clinical trials would be needed to clarify the prognostic role of cyNx in synchronous BMRCC. Furthermore, the positive combination of good KPS, low cerebral burden of disease, surgical resectability and the absence of other metastatic sites, as predictors of benefit from cyNx, should be prospectively confirmed.

## 5. Current Treatment Strategies

Once BMRCC is diagnosed, it is mandatory to identify the correct multimodal therapeutic strategy with the ultimate aim of improving patients’ survival. To date, the treatment strategy for BMRCC is usually tailored to patient’s characteristics, which resembles that of mRCC as a whole, but no general consensus on an ‘ideal’ strategy has been reached so far.

Beyond the usual (but not necessarily still valid) beliefs that the blood brain barrier (BBB) may reduce the availability of many oncologic drugs in CNS, and that BMRCC are radioresistant, the major issue is that patients with BM are almost always excluded from clinical trials, leading to a lack of high-level evidence in support of any given treatment [36].

On the other hand, much evidence has shown that the BBB is dynamic and that its permeability varies concomitantly with the the functional requirements of signaling systems in the brain. For instance, the tight junctions of the BBB can be disrupted in the presence of cerebral oedema, and due to the effect of pro-inflammatory cytokines, such as interferon-γ and tumor necrosis factor-α, other than physically disrupted after tumor extension. These observations derive from histopathological findings in brain tumor series, which have demonstrated the presence of infiltrating immune cells even in primary brain tumors specimens, macrophages but also CD4+ and CD8+ lymphocytes, as well as dynamic markers of the immune response such as PD-L1. Taken together, these factors demonstrate that the BBB is a relative, rather than an absolute barrier, when considering implications for the trafficking of immune cells or the delivery of cancer therapeutics [37].

Hence, due to the apparent controversy on BM characteristics and treatment efficacy, we discuss the principal data published in the literature regarding the various therapeutic approaches for BMRCC with the aim of identifying the most effective strategies.

### 5.1. Surgical Resection

A neurosurgical approach improves outcome when feasible, and especially in the case of single BMRCC. Usually, surgical resection is chosen for the following reasons: the propensity of BMRCC to bleed (46% of incidence), which is indeed a potentially life-threatening event, the frequent presence of intracranial edema, often causing neurological symptoms/impairment, and the anatomical/technical possibility of achieving a Gross Total Resection (GTR), i.e., the radical exeresis of BM.

Neurosurgeons consider some well-defined clinico-prognostic criteria during the operability-defining process in case of elective procedures: KPS, number of BMs (more than 50% of patients with cerebral metastases from RCC present multiple lesions), BM site, as well as the presence of extracranial metastasis [13].

Interestingly, many BM from RCC are well-circumscribed and relatively firm, and thus, suitable for surgical resection [38].

Moreover, it has been postulated that a long interval between CNS metastases occurrence and the diagnosis of the primary tumor represents an indicator of better prognosis. Therefore, an aggressive treatment must be considered in patients presenting with delayed CNS metastases, since they usually show a relatively long survival and good prognosis [14]

### 5.2. The Role of Radiotherapy

Whole brain radiotherapy (WBRT) after a radical resection of BMRCC is usually not suggested.

In the case of residual disease after surgery, or of non-surgical disease, Gamma Knife radiosurgery (GKS) may provide a more effective tumor control [27,39,40,41], as compared to WBRT.

Although the rate of complete response of BMRCC after GKS is relatively low, growth control is high (nearly 85%) and the dose delivered to the tumor margins significantly correlated with the control of peritumoral oedema [42].

To date, GKS is effective even in patients with multiple metastases, and repeated GKS sessions can be proposed for newly developed brain metastases

Predictive factors positively impacting OS after GSK are young age, good preoperative KPS, long interval from initial cancer diagnosis to brain metastasis, a dose of more than 20 Gy to the tumor margins, and maximal treatment dose [3].

Moreover, in order to establish a risk score predictor of brain recurrence after GSK, Rades et al. analyzed the clinico-radiological characteristics of 45 BMRCC patients who underwent GSK, and identified two groups of patients with significantly different risk of recurrence. In particular, patients with multiple brain metastasis (>3), with an infratentorial location, were considered at high risk of recurrence outside the irradiated area [43].

Further analyses were conducted with the aim of assessing the safety and efficacy of concomitant targeted therapies (TT). In particular, vascular endothelial growth factor receptors tyrosine kinase inhibitors (VEGFR-TKIs) were assessed, and it was concluded that stereotactic radiosurgery (SRS) is highly effective in patients with BMRCC and the association with TT does not yield a higher risk of n. Furthermore, in multivariate analysis, a minimal dose > 17 Gy and the administration of a concomitant TKI were both associated with higher local control [44].

### 5.3. TKIs

The main obstacle to the efficacy of anti-cancer drugs in CNS is represented by the BBB and to a lesser extent of the blood–tumor barrier (BTB) that could unpredictably impair drug penetration into CNS [45,46]. Many series showed a significant activity of TT on BM, suggesting that these drugs may succeed in overcoming BBB and BTB. In particular, anti-VEGFR TKIs are not associated with an increased frequency of intracerebral bleeding, underlying a safety use in BMRCC [47,48]. Moreover, if VEGFR-TKIs-induce hypertension can be well-controlled [49].

In this regard, the PREDICT (Patient characteristics in RCC and Daily practICe Treatment with sorafenib) trial evidenced the safety and efficacy of sorafenib in BMRCC [50].

Interestingly enough, a retrospective sub-analysis of the phase III Treatment Approaches in Renal Cancer Global Evaluation Trial (TARGET) study evaluated the incidence of brain metastases in a subgroup of patients with metastatic renal cell carcinoma (RCC), who were randomly assigned to receive sorafenib or placebo. The overall incidence of newly occurring brain metastases in metastatic patients (without BM at baseline) receiving sorafenib was 3% compared with 12% in patients receiving placebo (*p* < 0.05), a difference that was maintained after 1 (*p* = 0.0447) and 2 years (*p* = 0.005) of treatment compared with the placebo group. Therefore, this suggests that sorafenib may have some preventive effect toward the development of the new brain metastases [51]. Another study explored this putative protective role of TKIs (Sorafenib of Sunitinib) from the development of BMs, and concluded that treatment with TKIs reduces the incidence of brain metastasis in mRCC. The authors found that mOS was longer in the TKI-treated group (25 months vs. 12.1 months, *p* < 0.0001) and the 5-year rate of brain metastasis development was 40%, and 17%, respectively (non-TKI pre-treated group vs. TKI pre-treated group) and TKIs treatment was associated with lower incidence of brain metastasis in Cox multivariate analysis [52].

Furthermore, Gore and coll. analyzed the activity of sunitinib on BMRCC within the global expanded acces program (EAP), and found that 9% of patients achieved objective response rate (ORR), while 33% had a stable disease (SD) for more than 3 months, with an overall 3-month or more clinical benefit of 42% [53,54,55]. In the Italian cohort of patients with BMRCC who received sunitinib within the same EAP, the ORR was 4% with 35% of patients with a SD of at least 3 months, and a clinical benefit of 39% [56].

Regarding pazopanib, some studies showed an efficacy on BMRCC, with a SD in up to 60% of patients, and a 13% BM shrinkage [57,58,59,60].

Negrier et coll. published two case-reports regarding the safety and activity of Cabozantinib in BMRCC refractory to SRS, suggesting an ability to reach CNS and generate BM shrinkage [61] and, recently, Hirsch L. et coll confirmed, in a retrospective analysis, the efficacy of Caboantinib with respect to BMRCC [62]

As far as other TT is concerned, the RECORD1 and the REACT studies showed the safety of the mTOR inhibitor everolimus in patients with BMRCC previously treated with antiangiogenetic drugs [63,64], while the ARCC (Global Advanced Renal Cell Carcinoma) trial demonstrated temsirolimus safety in previously resected BMRCC or in patients who previously underwent radiotherapy without neurological deficits [65].

### 5.4. Immune Checkpoint Inhibitors (ICIs)

ICIs revolutionized the treatment landscape of RCC, but BMRCC patients were commonly excluded from trials. As a consequence, safety and efficacy data in this subgroup of patients derive from small series, which overall have shown comparable efficacy to that evidence within pivotal trials [66].

The biological rationale of the use of ICIs in case of BMRCC relies on the concept that inflammatory microenvironment of BMRCC is highly immunogenic, as evidenced by the pronounced infiltration of tumor infiltrated lymphocytes (TIL) [67].

Nevertheless, BM RCC, often characterized by high levels of CD8 + TIL, present a high amount of Programmed cell-death 1 (PD-1) that may be functionally impaired in the presence of programmed cell-death ligand-1 (PD-L1), even if the survival is not correlated with TILs number or PDL1 expression. In fact, Zhang et coll. evidenced that PD-1, PD-L1, and PD-L2 were differentially expressed between the primary and metastatic tumors, and PD-1 is poorly expressed in BM (nearly 10%) [68].

Further analysis in this regard were conducted by Derosa L. et. al., who compared PD-L1 and MET expression assessed by immunohistochemistry in both primary tumor (considering tumor cells and immune cells for PDL-1) and BM. The authors demonstrated the inter-tumor heterogeneity of both PDL-1 and MET expression. In particular, the discordance rate in PD-L1 for tumor cells, immune cells and MET between the primary tumor and BM were 40%, 22%, and 67%, respectively, suggesting that the assessment as predictive biomarkers probably require analysis of metastatic lesions [69].

In relation to the clinically available data, the NIVOREN trial demonstrated that, out of 55 BM patients (67% of whom were not treated for their BM), 60% achieved a 3-months PFS after the treatment with Nivolumab. Furthermore, among 44 patients assessed for BM response, ORR was 23%, while local progressive disease was 48%. Finally, neurologic deterioration requiring steroids was observed in 32% of patients [70].

Furthermore, the Italian expanded access program reported on a large series (389 patients) of BMRCC treated with nivolumab beyond first-line, including 32 patients with BM that did not require radiotherapy or steroids [71]. The results showed that the response rates were irrespective of age, histology, previous lines of therapy, and the presence of brain or bone metastasis. For BM patients, 6-month and 12-month OS were 87%, and 66.8%, respectively [72].

Another unsolved issue is the safety and efficacy of the combination of radiotherapy and ICIs, and the most appropriate timing for the utilization of these two different strategies. To date, the combination of ICIs and radiotherapy finds the rationale in the delayed response that characterizes immunotherapy treatment and the consequent need for a rapid antitumor activity especially in symptomatic BM patients [73,74].

To date, there is a lack of data regarding both the optimal timing of ICIs with radiotherapy, the optimal fractionation scheme of radiotherapy. The major difficulty was in overcoming the need for corticosteroid administration of patients treated with SRT or WBRT, which is hypothesized to potentially antagonize the effect of ICIs [75]. In this regard, Garant et al. published a systematic review concerning the role of the concomitant use of corticosteroids and ICIs, and demonstrated that only 20% of the studies showed a detrimental effect of the concomitant use of corticosteroids during immunotherapy. Moreover, there was no objective data, either on the type of corticosteroids or about the dose threshold above which an interaction could be measured clinically [76]. The activity and efficacy of systemic therapies in BMRCC are summarized in Table 2.

### 5.5. Ongoing Clinical Trials

More recently, a number of clinical trials have been designed with the aim of overcoming the paucity of data regarding the correct treatment strategies in this peculiar setting.

Among the most interesting ongoing clinical trials, NCT03967522 is a multicenter, open-label, exploratory, single-arm, prospective phase II study with the aim of assessing the efficacy and safety of cabozantinib in patients with BMRCC [77].

Furthermore, NCT04187872 is an open-label, historically controlled pilot study investigating the immune effect of Laser Interstitial ThermotHerapy (LITT), that is a a laser catheter implanted into the tumor and heating it to temperatures high enough to kill the tumor cells, with the concomitant administration of pembrolizumab in adult patients with a primary cancer. The trial has been approved by the FDA for treatment of recurrent brain metastasis after prior SRS [78].

Finally, NCT04434560 is a phase 2 study designed to assess the feasibility and efficacy of neoadjuvant immunotherapy (ipilimumab and nivolumab) in patients with previously untreated, surgically-resectable, solid tumor brain metastases [79].

## 6. General Conclusions and Future Perspectives

BM occur in RCC patients with an incidence that is now around 10%. The prognosis is poor, being mOS around 10 months. Prognostic factors are KPS, the number of BM, the presence of a sarcomatoid component and the existance of extracranial metastases.

In relation to treatments options, while BM patients are usually excluded from clinical trials and no validated strategies are universally agreed upon, different treatment options such as radiosurgery and surgery can be applied according to the location and the number of metastatic tumors in the brain. Local treatments, such as surgery or radiosurgery are indeed crucial in patients with symptomatic CNS involvement, also because the relatively high tendency of bleeding of BMRCC due to their exasperated vascular network. In relation to systemic treatment, TT or ICIs agents proved to be active and safe. Although, the limited amount of available data and their mainly retrospective nature prevents any definitive conclusions on the efficacy.

This review underly the existence of a subgroup of patients that, perhaps incorrectly, have been not considered as an intermediate prognostic subgroup. Radiological advances in images, and the knowledge of BBB and brain tumor microenvironment, will help to more adequately define the therapeutic strategy for these patients.

Hopefully, ongoing trials will clarify this important therapeutic issue.

## Figures and Tables

**Figure 1 cancers-13-02114-f001:**
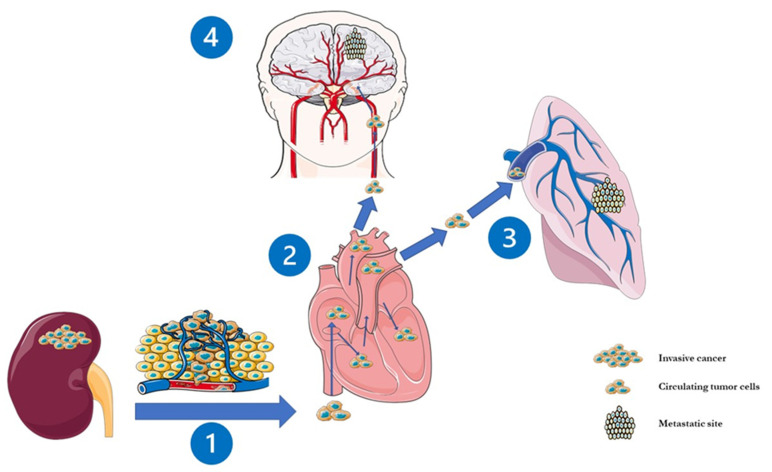
The cava-pathway and subsequent steps in the development of brain metastasis in renal cell carcinoma. The metastatic spread seems to begin with RCC-derived microvesicles CD105+ which break off from the primary tumour site and disseminate through the cava-type hematogenous pathway in 75% of the cases (**1**). Then, these microvesicles, carrying a cancer stem cell phenotype which stimulate angiogenesis, are transported through the right heart (**2**) into the pulmonary capillaries and (**3**) arterial circulation and finally attain the cerebral vasculature (**4**).

**Table 1 cancers-13-02114-t001:** Calculation of BMRCC prognostic scores.

DS-GPA *			
Points	0	1	2
KPS	<70	70–80	90–100
No of BMs	>3	2–3	1
**RTOG RPA**			
Class	I	II	III
KPS	≥70	≥70	<70
	and	and	and
Age	<65	all	all
	and	and	and
Extracranial metastases	No	No/Yes	No/Yes
**BS-BM ***			
Points	0	1	
KPS	≤70	≥80	
Systemic disease	PD	SD-PR-CR-NED	
Extracranial metastases	Yes	No	
**CERENAL ***			
Points	0	1	
KPS	>70	≤70	
Age	≤50	>50	
PD of systemic disease	No	Yes	
Extracranial metastases	No	Yes	
No of BM	1	≥2	
SRS	Yes	No	

* Final score is obtained by sum of all individual points; GPA of 4.0 indicates best prognosis and 0.0 indicates worst; for BS-BM and CERENAL, cut-off of worse prognosis was taken at ≤2 and ≤4, respectively.

**Table 2 cancers-13-02114-t002:** Effect of systemic therapy (anti-VEGFR-TKIs and ICIs) on BMRCC.

Authors	Drug	Previous Treatment	Main Results
*TKIs*			
Jäger et al., 2015 (PREDICT trial)	Sorafenib	37% of patients received sorafenib in first-line, while 63% were pre-treated with cytokines or TT (sunitinib, temsirolimus, pazopanib, or bevacizumab plus interferon).	Safety and efficacy of Sorafenib in BMRCC was confirmed. The median duration of sorafenib in the BM group was 7.3 months.
Gore et al., 2015 (Sunitinib global EAP)	Sunitinib	12% treatment-naïve; 10% previously treated with VEGFR-TKIs; 68% previously treated with cytokines	9% of BMRCC patients achieved an objective response rate, 33% had stable disease (SD) for more than 3 months, leading to an overall clinical benefit of 42%
Jacobs et al. 2013; Matrana et al., 2013; Roberto et al. 2015; Santoni et al. 2015	Pazopanib	At list one previous treatment (range, 1–5) including all prior TT, cytokines, cytotoxics, and other experimental agents.	SD observed in 60% of patients, with a percentage of regression of brain lesions in 13% of cases
Negrier et al., 2018	Cabozantinib	Anti VEGFr-TKI, anti-PD1 treatment and SRS	Cabozantinib proved able to reach the brain and to induce regressions of BMRCC that were resistant to radiation and previous angiogenic VEGFR-TKIs (warning: just case reports)
ICIs			
Escudier et al., 2017 (NIVOREN trial); De Giorgi et al., 2019;Nivolumab Italian EAP)	Nivolumab Nivolumab	Previous anti-VEGFR-TKIs and/or cytokines; 67% of pts had not received local therapy for BM Anti-VEGFR-TKIs and/or cytokines	60% of pts achieved a PFS of at least 3 months; among 44 patients assessed for BM response, ORR was 23%, while local progressive disease was 48%. Neurologic deterioration requiring steroids was observed in 32% of patients OS and ORR in 32 BMRCC patients were not different from those of the overall study population

## Data Availability

No new data were created or analyzed in this study. Data sharing is not applicable to this article.

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
