# Peer review of "Prognostic Factors and Current Treatment Strategies for Renal Cell Carcinoma Metastatic to the Brain: An Overview"

_cancers, 2021, doi:10.3390/cancers13092114_

Round 1

Reviewer 1 Report

The authors have written a comprehensive and up to date review about renal cell carcinoma brain metastases, touching pathogenesis to accepted and upcoming treatments. This will be a useful read for oncologists and neurosurgeons alike. I have no modification to suggest.

Author Response

thank you

Reviewer 2 Report

The authors represent data of brain metastasis in kidney cancer patients. The originality and novelty of the study is very high. I can note that the study adds to the existing knowledge-base. The title reflects the content of the paper. It has a logical construction and is written in a clear and easily understandable style. The design of the study is consistent with its aims. The methods are modern and clearly described at all.

On my point of view, the conclusions are clear set up. The discussion section is not critique. It is not discussed the methodology used in the paper.

The references are accurate, up to date, and relevant. However, the paper could be accepted and published.

Author Response

thank you for your suggestions. we modified the conclusions may be in a more adequate matter (line 398 to 414)

Reviewer 3 Report

In this review, the authors described BMRCC prognostic factors and current treatment strategy. This is well written and the information would be helpful. The following points should be revised and clarified to improve the manuscript.

comments

  1. The abbreviation of SRS should be defined at its first appearance in the Abstract.
  2. In “4. Role of cytoreductive nephrectomy” section, the authors described prospective clinical trial would be needed to clarity the prognostic role of cyNx in BMRCC. The authors also described some positive data of cyNx. However, there are no information in terms of adverse event. I recommend the authors to add the information regarding cyNx adverse event.

Author Response

  1. done
  2.  we added the possible complications of cyNx (lines 209 to 212)